# FTFT: Efficient and Robust Fine-Tuning by TransFerring Training Dynamics

## Abstract

Despite the massive success of fine-tuning large Pre-trained Language Models (PLMs) on a wide range of Natural Language Processing (NLP) tasks, they remain susceptible to out-of-distribution (OOD) and adversarial inputs. Data map (DM) is a simple yet effective dual-model approach that enhances the robustness of fine-tuned PLMs, which involves fine-tuning a model on the original training set (i.e. reference model), selecting a specified fraction of important training examples according to the training dynamics of the reference model, and fine-tuning the same PLM on these selected examples (i.e. main model). However, it suffers from the drawback of requiring fine-tuning the same model twice, which is computationally expensive for large models. In this paper, we first show that 1) training dynamics are highly transferable across different model sizes and different pre-training methods, and that 2) main models fine-tuned using DM learn faster than when using conventional Empirical Risk Minimization (ERM). Building on these observations, we propose a novel fine-tuning approach based on the DM method: Fine-Tuning by transFerring Training dynamics (FTFT). Compared with DM, FTFT uses more efficient reference models and fewer training steps. Our experiments show that FTFT achieves better generalization robustness than ERM while spending less than half of the training cost.

## 1 Introduction

Current state-of-the-art performance in Natural Language Processing (NLP) is dominated by large, pretrained language models (PLMs), which are typically fine-tuned for downstream tasks. Scaling laws (Kaplan et al., 2020; Hoffmann et al., 2022) suggest that better downstream performance is achieved with larger pretrained language models. However, fine-tuning large PLMs is also more expensive than fine-tuning small PLMs, in terms of both computational resources and carbon emission (Strubell et al., 2019; Wu et al., 2022).

Moreover, despite making impressive progress on regular benchmarks, many studies have shown that fine-tuned PLMs lack robustness against out-of-distribution (OOD) and adversarial inputs. For instance, human annotators can easily exploit the weaknesses of fine-tuned PLMs to trick these models to yield incorrect predictions, on tasks such as Natural Language Inference (NLI) (Nie et al., 2020) and Hate Speech Detection (HSD) (Vidgen et al., 2021b).

The problem of robustness can be mitigated using dual-model approaches. With such approaches, first a **reference model** is trained to estimate the importance of each training instance, and then a **main model** is trained based on the outputs of the reference model (Nam et al., 2020; Utama et al., 2020; Sanh et al., 2021; Karimi Mahabadi et al., 2020; Zhang et al., 2022; Liu et al., 2021). In particular, the approach proposed by Swayamdipta et al. (2020) is attractive in view of its simplicity and the demonstration that it consistently improves model performance on OOD test datasets. First, a Data Map (DM) is constructed, based on the **training dynamics** from an initial fine-tuning run of the model (i.e., the reference model) on the full dataset. The DM method divides the training data into three subsets: ambiguous, hard-to-learn, and easy instances. The main model is then fine-tuned using only either the ambiguous or the hard-to-learn subset. In the standard DM approach, the reference model and the main model are the same PLM (e.g., DeBERTaV3$_{\text{Large}}$, He et al., 2023). However, a major drawback of this approach is that it improves robustness at the expense of ef-

ficiency, which is especially problematic when using large PLMs, because this approach requires fine-tuning the same model twice.

In this paper, we jointly address the issues of robustness and efficiency without sacrificing the properties that make DMs attractive. We achieve this by exploiting the transferability of training dynamics. Specifically, we study whether training instances identified by more efficient reference models can be used to improve the robustness of more capable — and often larger — main models. We make three key observations. First, *training dynamics are highly transferable* across different model sizes (e.g., DeBERTaV3$_{\text{Small}}$ as the reference model and DeBERTaV3$_{\text{Large}}$ as the main model) and different pre-training methods (PTMs), e.g., ELECTRA (Clark et al., 2020) as the reference model and DeBERTaV3 as the main model. Second, the condition for this transfer to work well is that reference models should be reasonably strong. Crucially, we identify a key property of effective reference models, namely that they typically identify higher ratios of training instances as easy cases, compared to less effective reference models. This observation can help us inspect whether a reference model would work well without training the main model. Third, the main model in the DM method learns faster than ERM, achieving good performance using fewer steps.

Based on our observations, we propose **Fine-Tuning by transFerring Training dynamics (FTFT)** to improve both the efficiency and the robustness of fine-tuning. Concretely, FTFT improves the efficiency of the DM method in two ways. First, FTFT uses more efficient reference models for identifying the subset of ambiguous training instances. Second, when using this data subset to fine-tune a more capable main model, FTFT uses substantially fewer training steps than ERM fine-tuning. Experimental results on two tasks, NLI and HSD, using two models, DeBERTaV3 (He et al., 2021) and ELECTRA (Clark et al., 2020), show that FTFT can achieve better generalization robustness than ERM, while lowering the training cost by a factor of more than two.

## 2 BACKGROUND

### 2.1 IMPROVING MODEL ROBUSTNESS USING DUAL-MODEL APPROACHES

Many previous studies have proposed dual-model approaches to improve model robustness. Nam et al. (2020) first train a reference model using generalized cross-entropy loss. Then, they train a main model while assigning higher weights to training instances that are hard for the reference model. Utama et al. (2020) use the same rationale, except for using a model that is trained on a random subset of the full training data as the reference model, as well as adding a confidence regularization mechanism. Sanh et al. (2021) use a Product-of-Expert (PoE) approach, by first training a reference model with limited capacity to capture dataset biases, and then training the main model to avoid these biases using PoE loss. Karimi Mahabadi et al. (2020) also adopt a PoE loss, but they train both the reference model and the main model in an end-to-end fashion. Zhang et al. (2022) investigate the idea of regularizing hidden representations by first training a reference model using ERM, and then using contrastive learning to make the hidden representations of the same class and different classes more similar and more separable. Liu et al. (2021) first train a less capable reference model using heavy regularization and vanilla SGD, and up-weigh the training instances that the reference model mis-predicts when training the main model. The DM method proposed by Swayamdipta et al. (2020) is a more nuanced approach to a similar idea. Instead of relying only on the correctness of the reference model, they use training dynamics to categorize training instances, and only use a subset of data to train the main model. We turn to the details of this method below.

### 2.2 DATA MAP

Swayamdipta et al. (2020) propose a dual-model approach for improving model robustness. First, a reference model is trained on the full original dataset. Then, a Data Map (DM) is built based on the observed training dynamics, by tracking the prediction probabilities of the true class ($p_{\text{true}}$) of each training instance across different epochs. Using the DM, training instances can be categorized into three groups: *ambiguous* (i.e. the standard deviation of $p_{\text{true}}$ is in the top $q\%$ of all training instances); *hard-to-learn* (i.e. the mean of $p_{\text{true}}$ is at the bottom $q\%$ of all training instances); and *easy* (i.e. neither ambiguous nor hard-to-learn). The threshold $q\%$ is fixed and typically set to $33\%$. Note that *ambiguous and hard-to-learn are not mutually exclusive*: a training instance can be categorized as both hard-to-learn (the average of its $p_{\text{true}}$ values is at the bottom $q\%$) and ambiguous

(the standard deviation of its $p_{\text{true}}$ values is among the top $q\%$). Finally, the main model is fine-tuned only on the $q\%$ most ambiguous or hard-to-learn datapoints. Swayamdipta et al. (2020) show that, with a slight loss of In-Distribution (ID) performance, this approach improves model performance on challenging Out-Of-Distribution (OOD) datasets. They also observe that training on ambiguous data leads to better performance than training on hard-to-learn data. We therefore mainly focus on ambiguous data, although we also study the interaction between ambiguous and hard-to-learn data.

Swayamdipta et al. (2020) uses the same PLM as both the reference and the main model. In contrast, Sar-Shalom & Schwartz (2023) show that a DM constructed by ELECTRA$_{\text{Large}}$ can be used to improve the robustness of DeBERTaV3$_{\text{Large}}$. However, they use a different approach: instead of training only on the $q\%$ most ambiguous datapoints, they add $k$ copies of this subset to the original training set; also, they do not investigate DM transferability further. Inspired by their observation, we study whether exploiting the transferability of training dynamics can help improve the efficiency of the DM method, while retaining the advantage of improved robustness.

## 3 EXPERIMENTAL SETUP

In our experiments we study the transferability of training dynamics on two tasks, Natural Language Inference (NLI) and Hate Speech Detection (HSD). We consider both ID and OOD performance. As a baseline, we also experiment with random DM (i.e., randomly selecting $q\%$ of the training data). Unless otherwise specified, we set the ratio $q\%$ of all DM methods to 33%.

**Data** To study the impact on model robustness, for each task, besides the original train set and ID validation set, we also include a few challenging OOD test sets.

For NLI, we use the MultiNLI dataset (Williams et al., 2018) as the train and ID validation set, because of its wide coverage of diverse inputs (10 genres). As OOD test sets, we use two challenging datasets designed to target weaknesses of models trained on MultiNLI: WANLI (Liu et al., 2022) and AdversarialNLI (Nie et al., 2020), which consists of three rounds of adversarial data collection.

For HSD, we use CAD (Vidgen et al., 2021a) as the train and ID validation set. CAD consists of Reddit posts covering diverse topics and writing styles, annotated for multiple categories. We frame the task as a binary classification task (hateful vs. non-hateful). Following Ramponi & Tonelli (2022), we mark identity-related abuse as hateful and all other categories as non-hateful. As OOD test sets, we use DynaHate (Vidgen et al., 2021b), which also contains three rounds of adversarial data collection, as well as a perturbed version for each round. Conceptualizations of hate speech vary widely across hate speech datasets. We selected these two datasets, because they use similar definitions and taxonomies concerning hate speech.

**Models** We focus on two PLMs, DeBERTaV3 (He et al., 2023) and ELECTRA (Clark et al., 2020), for two reasons. First, they have both been shown to perform well on text classification tasks. Second, they are available in different sizes (small, base, and large), allowing us to study the transferability between different model sizes. We also experiment with TinyBERT (Turc et al., 2020), an efficient but weak PLM, to study the impact of low reference model capacity, along with BERT (Devlin et al., 2019) and RoBERTa (Liu et al., 2019) to study the transferability across pretraining methods. We provide the cost for fine-tuning different PLMs in Appendix A.1.[1]

**Training** For NLI, we train all models for 60k steps ($\sim$ 5 epochs) and evaluate every 4k steps. For HSD, we train all models for 6k steps ($\sim$ 10 epochs) and evaluate every 400 steps. For all models, we use four different random seeds and report their average performance. Full training details (e.g., optimization, software and hardware) are included in Appendix A.2.

---

[1]We report PFLOPs rather than GPU hours because on our setup, which uses NVIDIA A100 GPUs, we've noticed occasional low GPU utilization, especially during the fine-tuning of smaller models such as TinyBERT. In such cases, reporting GPU hours would not reflect the computational costs accurately.

## 4 TRANSFERABILITY OF TRAINING DYNAMICS

In this section, we study the transferability of training dynamics in the DM method, i.e., whether we can use different reference and main models while maintaining the robustness of the main model. Specifically, we study whether training dynamics are transferable across different model sizes (§4.1, e.g., from DeBERTaV3$_{Small}$ to DeBERTaV3$_{Large}$) and pretraining methods (§4.2, e.g., from ELECTRA$_{Large}$ to DeBERTaV3$_{Large}$), for two reasons. First, transferability across model sizes can help improve the efficiency of constructing a DM, by enabling the use of more efficient reference models. Second, transferability across pretraining methods can help achieve such an efficiency gain, when more efficient versions of the main model pretraining method are either not available, or are not able to model the task sufficiently well. Moreover, transferability across different models can help us gain a better understanding of data ambiguity. Indeed, if the same subset of training instances is consistently identified as ambiguous by different reference models, this would suggest that ambiguity is to some extent intrinsic to data, rather than being completely model-dependent.

Our results show that training dynamics are transferable across different model sizes and pretraining methods, with a few exceptions. To understand the conditions for successful transfers, we analyze these few failure cases. We find that for successful transfer, reference models need to be reasonably strong in identifying easy training instances (§4.3). This finding can serve as a guideline for choosing reference models without training the main model, which can be computationally expensive.

### 4.1 TRANSFERABILITY ACROSS DIFFERENT MODEL SIZES

In this section, we study whether smaller and more efficient models can be used as reference models for training larger main models, without compromising robustness. For example, when using DeBERTaV3$_{Large}$ as the main model, we investigate whether there is a comparable performance when using DeBERTaV3$_{Small}$ versus using DeBERTaV3$_{Large}$ itself as reference models. Successful transfers of this type enables the use of more efficient reference models.

We show the results for DeBERTaV3 in Table 1 (across different model sizes) and make three observations. First, performance is better with larger PLMs as main models, even when we use ERM. In support of this claim, we note that DeBERTaV3$_{Large}$ performs better than DeBERTaV3$_{Small}$ and DeBERTaV3$_{Base}$ when fine-tuned with ERM or DM. This is the case for all test datasets, except when the reference model is trained on a random 33% subset of the train data. Second, by comparing DeBERTaV3$_{Large}$ fine-tuned with ERM and DM, we observe that ERM performs better on ID datasets, while DM outperforms ERM on OOD datasets. This is consistent with Swayamdipta et al. (2020). Third and most importantly, *training dynamics are transferable across different model sizes*: When using DeBERTaV3$_{Large}$ as the main model, changing the reference model from DeBERTaV3$_{Large}$ to DeBERTaV3$_{Small}$ or DeBERTaV3$_{Base}$ yields comparable performance.

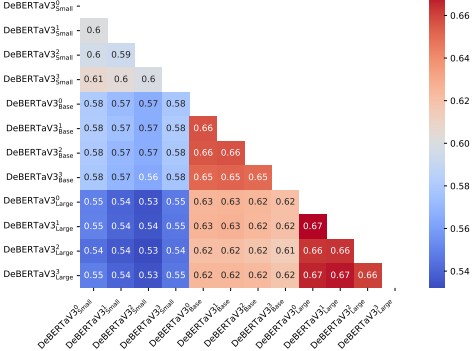

Figure 1: Consistency across different sizes of DeBERTaV3 on NLI. The numbers are the fractions (0-1) of the ambiguous training instances shared by two models. Training dynamics are transferable across different sizes: the fractions of shared ambiguous instances between models of different sizes are only slightly smaller than those between models of the same size but different random seeds (shown as superscript).

To understand this transferability, we analyze whether reference models of different sizes are consistent in identifying ambiguous training instances. Figure 1 displays the fraction of the ambiguous instances shared by reference models of different sizes, or of the same size but different random seeds. Consistent with our previous observation, the fractions of shared ambiguous instances between different sizes are only slightly smaller than those between the same size but different random seeds (for comparison, the fraction between random DMs is expected to be 0.33, since we select 33% most ambiguous instances).

| Mode | Main Model | Ref. Model | MultiNLI | WANLI | AdversarialNLI | | |
|------|-----------|-----------|----------|-------|------|------|------|
| | | | | | R1 | R2 | R3 |
| ERM | DeBERTaV3$_{Small}$ | - | 87.57 | 61.62 | 33.55 | 30.63 | 32.40 |
| ERM | DeBERTaV3$_{Base}$ | - | 90.00 | 64.61 | 43.55 | 33.23 | 34.15 |
| ERM | DeBERTaV3$_{Large}$ | - | **91.06** | 66.46 | 58.18 | 45.58 | 41.34 |
| DM | DeBERTaV3$_{Large}$ | Random | 90.74 | 65.31 | 53.30 | 42.03 | 38.60 |
| DM | DeBERTaV3$_{Large}$ | DeBERTaV3$_{Large}$ | 90.75 | 66.33 | 59.75 | 45.60 | 41.94 |
| | | Across different model sizes | | | | | |
| DM | DeBERTaV3$_{Large}$ | DeBERTaV3$_{Small}$ | 90.74 | 66.80 | 59.60 | 45.63 | 42.04 |
| DM | DeBERTaV3$_{Large}$ | DeBERTaV3$_{Base}$ | 90.52 | 66.61 | **61.43** | 46.73 | 41.58 |
| | | Across different pretraining methods | | | | | |
| DM | DeBERTaV3$_{Large}$ | ELECTRA$_{Small}$ | 90.91 | 62.09 | 49.62 | 38.5 | 35.98 |
| DM | DeBERTaV3$_{Large}$ | ELECTRA$_{Base}$ | 90.63 | 66.58 | 59.77 | 46.25 | 42.29 |
| DM | DeBERTaV3$_{Large}$ | ELECTRA$_{Large}$ | 90.80 | 66.42 | 58.95 | 44.57 | 41.52 |
| DM | DeBERTaV3$_{Large}$ | BERT$_{Large}$ | 90.03 | **66.89** | 60.40 | **47.30** | **43.71** |
| DM | DeBERTaV3$_{Large}$ | RoBERTa$_{Large}$ | 90.71 | 66.38 | 58.77 | 46.47 | 41.73 |

| Mode | Main Model | Ref. Model | CAD | DynaHate-Original | | | DynaHate-Perturb | | |
|------|-----------|-----------|-----|------|------|------|------|------|------|
| | | | | R2 | R3 | R4 | R2 | R3 | R4 |
| ERM | DeBERTaV3$_{Small}$ | - | 76.58 | 56.89 | 59.29 | 63.48 | 59.55 | 66.60 | 61.48 |
| ERM | DeBERTaV3$_{Base}$ | - | 78.64 | 60.53 | 64.28 | 68.89 | 60.81 | 69.48 | 63.12 |
| ERM | DeBERTaV3$_{Large}$ | - | **81.69** | 75.44 | 73.33 | 76.12 | 70.62 | 77.41 | 68.89 |
| DM | DeBERTaV3$_{Large}$ | Random | 76.23 | 63.38 | 61.59 | 71.21 | 64.05 | 72.10 | 62.88 |
| DM | DeBERTaV3$_{Large}$ | DeBERTaV3$_{Large}$ | 81.58 | 79.18 | 76.87 | 77.73 | 73.35 | 76.63 | 67.54 |
| | | Across different model sizes | | | | | | | |
| DM | DeBERTaV3$_{Large}$ | DeBERTaV3$_{Small}$ | 81.15 | 80.68 | **79.57** | **79.86** | 76.47 | 78.03 | 70.32 |
| DM | DeBERTaV3$_{Large}$ | DeBERTaV3$_{Base}$ | 80.12 | 80.13 | 76.35 | 78.82 | 74.60 | 77.82 | 68.68 |
| | | Across different pretraining methods | | | | | | | |
| DM | DeBERTaV3$_{Large}$ | ELECTRA$_{Small}$ | 79.74 | 78.09 | 77.40 | 78.75 | 75.26 | 76.79 | 70.05 |
| DM | DeBERTaV3$_{Large}$ | ELECTRA$_{Base}$ | 80.37 | **81.47** | 78.16 | 78.38 | **76.99** | **78.51** | **71.01** |
| DM | DeBERTaV3$_{Large}$ | ELECTRA$_{Large}$ | 79.48 | 76.71 | 75.97 | 78.58 | 73.55 | 77.80 | 69.81 |
| DM | DeBERTaV3$_{Large}$ | BERT$_{Large}$ | 79.75 | 79.10 | 75.87 | 77.64 | 72.94 | 77.10 | 67.47 |
| DM | DeBERTaV3$_{Large}$ | RoBERTa$_{Large}$ | 80.56 | 80.42 | 77.26 | 79.48 | 73.11 | 77.31 | 69.48 |

Table 1: Transferability across different model sizes and pretraining methods, using DeBERTaV3 as the main model on NLI (top, accuracy) and HSD (bottom, macro-F1). We compare the performance of 1) DeBERTaV3 of different sizes fine-tuned using ERM, 2) DeBERTaV3$_{Large}$ as the main model, using random DM (random 33% training instances), and DeBERTaV3$_{Large}$ as the reference model (Ref. Model) to construct DM (original DM), 3) DeBERTaV3$_{Large}$ as the main model, using DeBERTaV3$_{Small}$ and DeBERTaV3$_{Base}$ as reference models to construct DM, 4) DeBERTaV3$_{Large}$ as the main model, using ELECTRA of different sizes as reference models to construct DM. R1-R4 in AdversarialNLI and DynaHate refer to different rounds of collected data. Training dynamics are transferrable across different sizes and pretraining methods: DM methods using different reference model sizes and pretraining methods show comparable performance.

## 4.2 TRANSFERABILITY ACROSS DIFFERENT PRETRAINING METHODS

In this section, we study the transferability of training dynamics across different pretraining methods, by comparing performance when the reference and main models are trained using the same pretraining method, or using different pretraining methods. If transfers of this type are successful, we can improve the efficiency of DM methods in case there is no version of the main model pretraining method which is both efficient and effective as the reference model.

Table 1 shows the results when using DeBERTaV3$_{Large}$ as the main model with different reference models (across different pretraining methods). *In most settings, training dynamics are transferable*

*across different pretraining methods*. We note that in most cases DeBERTaV3$_{\text{Large}}$ achieves comparable performance when trained by transferring a DM from various reference models. However, such transfers are not always successful. When using ELECTRA$_{\text{Small}}$ as the reference model, the performance is clearly worse on the OOD datasets of NLI than using ERM on challenging datasets. We hypothesize that ELECTRA$_{\text{Small}}$ is not strong enough for constructing effective DMs for MultiNLI, and will analyze this further in §4.3.

## 4.3 HOW EFFICIENT CAN WE BE?

We have shown that training dynamics are transferable across different model sizes and pretraining methods. In this section, we study the conditions for successful transfers. Specifically, we focus on answering two questions: 1) whether we can use very efficient but weak models as reference models; and 2) what are the differences between effective and ineffective reference models. Answers to these questions can serve as guidelines for selecting efficient yet effective reference models.

**The Use of Efficient but Weak Models**    To answer this question, we compare the performance of a wide range of methods of three types. First, we consider four models fine-tuned with ERM: the small, base, and large versions of ELECTRA, and TinyBERT (Turc et al., 2020), which is a very weak but efficient PLM. We use these four models because they have different sizes and capabilities (ELECTRA$_{\text{Large}}$ > ELECTRA$_{\text{Base}}$ > ELECTRA$_{\text{Small}}$ > TinyBERT). Second, we use these models as reference models to fine-tune DeBERTaV3$_{\text{Large}}$ using the DM method. By using reference models with different capabilities to fine-tune the same main model, we can inspect the impact of reference model capability on transferability. Third, we also include the results with ELECTRA$_{\text{Large}}$ as the main model, and different sizes of ELECTRA as reference models. By comparing results with different main models, we can better understand whether successful transfer of training dynamics is due to the compatibility between reference and main models, or due to the capability of the reference model itself. We also include the results for ELECTRA$_{\text{Large}}$ fine-tuned with a random DM as a baseline. We show our results for NLI in Table 2 and include the results for HSD in Appendix C. We make two observations.

First, poorly performing reference models, such as TinyBERT and ELECTRA$_{\text{Small}}$ fine-tuned with ERM, lead to failed transfers. Also, the worse the reference models perform, the worse the OOD performance of their main models are, e.g., TinyBERT leads to worse main model performance than ELECTRA$_{\text{Small}}$. Moreover, the success of transfers mostly depends on the reference, rather than the main models: transfers from ELECTRA$_{\text{Small}}$ to both ELECTRA$_{\text{Large}}$ and DeBERTaV3$_{\text{Large}}$ are unsuccessful. Second, using weak reference models for DM does not negatively affect ID performance much. For instance, transfers from ELECTRA$_{\text{Small}}$ to DeBERTaV3$_{\text{Large}}$ yields the best accuracy on MultiNLI. We suspect the reason is that, weak models usually identify easy training instances as ambiguous data, and easy training instances have been found to be sufficient for obtaining satisfactory ID performance (Swayamdipta et al., 2020).

**Differences between Effective and Ineffective Reference Models**    Because some reference models lead to failed transfers, it is important to understand the differences between effective and ineffective reference models. To answer this question, we start by considering the differences between a weak and a reasonably strong reference model when categorizing training data.

Assume we have a weak reference model which is only able to fit the easy training instances but fails to fit hard training instances. As a result, this weak reference model will assign increasing $p_{\text{true}}$ to easy training instances across different epochs, while keeping $p_{\text{true}}$ for hard training instances around the values expected in a random guessing scenario. This reference model behavior means that $p_{\text{true}}$ will exhibit high standard deviations on easy cases, which will therefore be identified as ambiguous data; while hard training instances will have lower means for $p_{\text{true}}$ and therefore be identified as hard-to-learn data. In contrast, a reasonably capable reference model can fit the easy part of the training instances during the early stage of training, which makes these instances have both high means and low standard deviations for $p_{\text{true}}$. In contrast, $p_{\text{true}}$ for harder instances will gradually increase across different epochs, making these instances yielding relatively low means and high standard deviations for $p_{\text{true}}$. As a result, such cases will be identified as both ambiguous and hard-to-learn (i.e. we expect a large overlap in these subsets). We validate this difference between weak and strong reference models, by comparing the changes of $p_{\text{true}}$ between hard-to-learn and other data

| Mode | Main Model | Ref. Model | MultiNLI | WANLI | AdversarialNLI | | |
| --- | --- | --- | --- | --- | --- | --- | --- |
| | | | | | R1 | R2 | R3 |
| ERM | TinyBERT | - | 67.32 | 43.40 | 23.30 | 28.10 | 30.94 |
| ERM | ELECTRA$_{Small}$ | - | 81.98 | 54.11 | 23.38 | 28.57 | 30.25 |
| ERM | ELECTRA$_{Base}$ | - | 88.53 | 63.06 | 34.58 | 30.73 | 31.29 |
| ERM | ELECTRA$_{Large}$ | - | 90.75 | 65.85 | 54.20 | 39.38 | 36.10 |
| DM | DeBERTaV3$_{Large}$ | TinyBERT | 89.17 | 60.02 | 41.83 | 34.58 | 34.54 |
| DM | DeBERTaV3$_{Large}$ | ELECTRA$_{Small}$ | **90.91** | 62.09 | 49.62 | 38.50 | 35.98 |
| DM | DeBERTaV3$_{Large}$ | ELECTRA$_{Base}$ | 90.63 | **66.58** | **59.77** | **46.25** | **42.29** |
| DM | DeBERTaV3$_{Large}$ | ELECTRA$_{Large}$ | 90.80 | 66.42 | 58.95 | 44.57 | 41.52 |
| DM | ELECTRA$_{Large}$ | ELECTRA$_{Small}$ | 89.88 | 61.53 | 45.90 | 36.20 | 31.89 |
| DM | ELECTRA$_{Large}$ | ELECTRA$_{Base}$ | 90.40 | 66.09 | 54.10 | 40.97 | 37.31 |
| DM | ELECTRA$_{Large}$ | ELECTRA$_{Large}$ | 90.33 | 65.37 | 53.73 | 39.67 | 36.17 |
| DM | ELECTRA$_{Large}$ | Random | 89.99 | 65.03 | 51.25 | 39.02 | 34.98 |

Table 2: Performance on NLI with conventional fine-tuning (ERM) and DM using the 33% most ambiguous data identified with different reference models. Random in Ref. Model means randomly selecting 33% of the train data. The rows marked in gray are the results for DM training where the transfer was not successful, and the corresponding reference models. Successful transfer requires the reference model to be reasonably strong: reference models of clearly worse performance lead to degraded OOD performance for the main models.

| | NLI: MultiNLI | | | HSD: CAD | | |
| --- | --- | --- | --- | --- | --- | --- |
| | 25% | 33% | 50% | 25% | 33% | 50% |
| ELECTRA$_{Small}$ | 55.91% | 47.77% | 35.84% | 68.72% | 62.10% | 46.07% |
| ELECTRA$_{Base}$ | 69.10% | 62.88% | 46.62% | 73.36% | 65.54% | 47.56% |
| ELECTRA$_{Large}$ | 66.91% | 59.08% | 45.00% | 67.80% | 60.03% | 45.01% |
| DeBERTaV3$_{Small}$ | 67.08% | 61.31% | 46.08% | 72.05% | 65.00% | 47.94% |
| DeBERTaV3$_{Base}$ | 71.39% | 64.02% | 46.90% | 73.20% | 65.80% | 48.04% |
| DeBERTaV3$_{Large}$ | 72.97% | 64.84% | 46.97% | 74.06% | 65.89% | 47.72% |
| TinyBERT | 51.29% | 38.44% | 20.01% | 63.58% | 54.45% | 40.20% |

Table 3: Differences between effective and ineffective reference models. We show the percentages of easy data identified by the data maps of different models; cells marked in gray are the ratios of easy data identified by ineffective reference models. The column names indicate different thresholds for $q\%$ in the DM method. The key difference between effective and ineffective reference models lies in their ability to identify easy cases: compared with other models, fewer data points are identified as easy by TinyBERT on both tasks, and by ELECTRA$_{Small}$ on NLI.

identified by them, and include the results in Appendix B. Because we select a fixed percentage $q\%$ of instances as ambiguous or hard-to-learn, the larger overlap results in a larger percentage of data instances to be classified as easy when using strong reference models.

This reasoning is supported by our previous observation on the ID performance of unsuccessful transfers (Table 2). Concretely, the fact that weaker reference models (e.g. TinyBERT) identify easy training instances as ambiguous, and that using only easy training instances for fine-tuning will produce comparable ID performance but degraded OOD performance (Swayamdipta et al., 2020), together explain our results for weak reference models.

To further validate our reasoning, we compute the percentages of easy training instances from different reference models and show the results in Table 3. In addition to the default $q\% = 33\%$, we also experiment with $q\% = 25\%$ and $q\% = 50\%$. Consistent with our reasoning, ineffective reference models (i.e. ELECTRA$_{Small}$ on NLI and TinyBERT on both datasets) indeed identify fewer data points as easy compared with other models. Furthermore, the overlap between hard-to-learn and ambiguous instances in successful transfers is usually very large. For example, with $q\% = 50\%$, all effective reference models identify more than 45% of the training data as easy, (the maximum is 50%, when ambiguous and hard-to-learn data align perfectly).

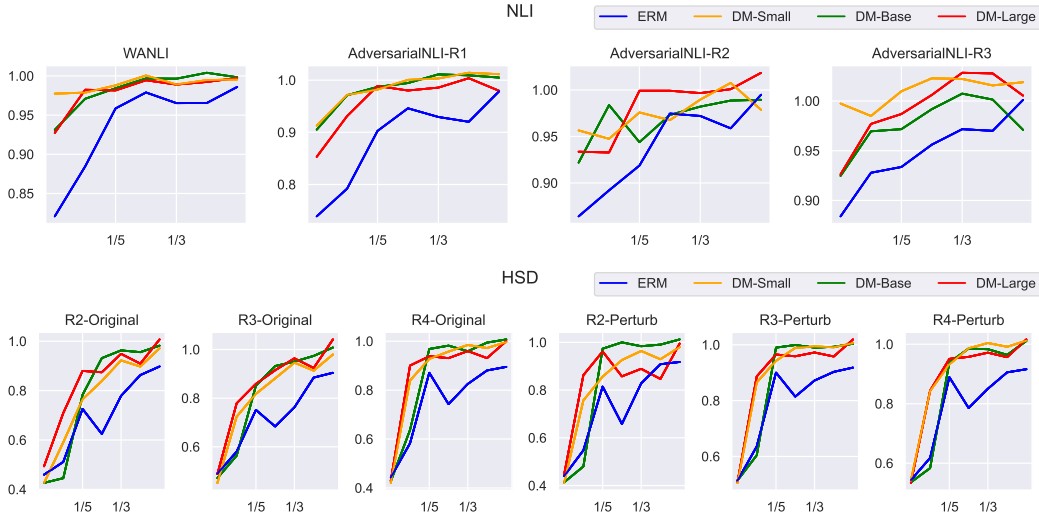

Figure 2: Performance on NLI (top) and HSD (bottom) when training the main model (ELECTRA) with fewer training steps. ERM is standard ERM fine-tuning on the full training set. DM-* refers to fine-tuning ELECTRA$_{\text{Large}}$ with the DM method, using reference model ELECTRA$_*$. The X-axis is the percentile of full training steps used, ranging from 1/15 to 7/15 of the total number of training steps. The Y-axis is the **percentage of performance compared with a model trained on the total number of training steps**. Fine-tuning the main model using data maps is much faster than ERM: the models achieve close-to-$100\%$ performance with only 1/3 of the training steps.

## 5 FTFT: EFFICIENT AND ROBUST FINE-TUNING BY TRANSFERRING TRAINING DYNAMICS

Given the transferability of training dynamics across model sizes and pretraining methods (§4), we can improve the efficiency of the DM method by using more efficient reference models. However, after training the reference model, DM method still requires fine-tuning the main model in the same way as ERM, making it less efficient than ERM. In this section, we investigate whether we can overcome this limitation: we show that models trained using DM learn faster than ERM, and fine-tuning the main model in DM for much fewer than the full training steps of ERM is already sufficient. From our observations, we propose a novel approach: Fine-Tuning by transFerring Training dynamics (FTFT). We show that FTFT consistently offers better efficiency and robustness over ERM.

**DM Trains Faster**   Figure 2 shows the OOD test performance of ELECTRA$_{\text{Large}}$ fine-tuned with fewer steps (i.e. from 1/15 to 7/15 of the total training steps). We compare DM with different reference model sizes against ERM. To better show the different learning speeds, instead of the absolute performance scores, we show the percentages compared with using the total number of training steps for each method. Clearly DM methods learns much faster than ERM on all OOD datasets: the performance curves of DM methods are almost always above those of ERM. We also observe that DM methods using only 1/3 of the total number of training steps already achieve comparable performance as using all training steps. This result suggests that we can further improve the efficiency of the DM method by training with fewer steps, while maintaining its robustness advantage over ERM. We show the results for DeBERTaV3$_{\text{Large}}$ in Appendix C and observe similar trends.

**FTFT: Achieving both Efficiency and Robustness**   FTFT involves two crucial changes to DM method. First, FTFT uses more efficient PLMs as reference models. Second, to improve the efficiency of the main model, FTFT uses only 1/3 of the training steps used for ERM. We choose 1/3 because we select 33% of the most ambiguous training instances, and this choice means we keep the same number of epochs as ERM. Nevertheless, we recommend to determine the number of training steps by monitoring model performance. Table 4 summarizes the performance of FTFT using DeBERTaV3$_{\text{Large}}$ as the main model, and DeBERTaV3$_{\text{Small}}$ and DeBERTaV3$_{\text{Base}}$ as reference

| Mode | Main Model | Ref. Model | Cost | MultiNLI | WANLI | AdversarialNLI | | |
| --- | --- | --- | --- | --- | --- | --- | --- | --- |
| | | | | | | R1 | R2 | R3 |
| ERM | DeBERTaV3$_{Large}$ | - | 32.0 | **91.06** | 66.46 | 58.17 | 45.57 | 41.34 |
| DM | DeBERTaV3$_{Large}$ | DeBERTaV3$_{Large}$ | 64.0 | 90.75 | 66.33 | 59.75 | 45.60 | 41.94 |
| JTT | DeBERTaV3$_{Large}$ | DeBERTaV3$_{Large}$ | 64.0 | 90.80 | 66.06 | 59.52 | 45.57 | 41.83 |
| PoE | DeBERTaV3$_{Large}$ | TinyBERT | 32.0 | 91.02 | **67.16** | 59.80 | 46.45 | 42.71 |
| FTFT | DeBERTaV3$_{Large}$ | DeBERTaV3$_{Small}$ | 15.2 | 90.12 | 66.42 | **60.30** | 45.75 | **43.66** |
| FTFT | DeBERTaV3$_{Large}$ | DeBERTaV3$_{Base}$ | 19.7 | 90.14 | 66.47 | 59.77 | **46.65** | 42.71 |

| Mode | Main Model | Ref. Model | CAD | DynaHate-Original | | | DynaHate-Perturb | | |
| --- | --- | --- | --- | --- | --- | --- | --- | --- | --- |
| | | | | R2 | R3 | R4 | R2 | R3 | R4 |
| ERM | DeBERTaV3$_{Large}$ | - | 81.69 | 75.44 | 73.32 | 76.12 | 70.62 | 77.41 | 68.89 |
| DM | DeBERTaV3$_{Large}$ | DeBERTaV3$_{Large}$ | 81.58 | 79.17 | 76.87 | 77.73 | 73.34 | 76.63 | 67.54 |
| JTT | DeBERTaV3$_{Large}$ | DeBERTaV3$_{Large}$ | 81.17 | 75.54 | 72.80 | 76.70 | 72.45 | 76.11 | 67.92 |
| PoE | DeBERTaV3$_{Large}$ | TinyBERT | **81.70** | 76.87 | 73.49 | 76.58 | 70.46 | 77.17 | 68.28 |
| FTFT | DeBERTaV3$_{Large}$ | DeBERTaV3$_{Small}$ | 80.73 | 78.77 | **77.53** | **79.48** | **76.19** | 77.53 | 69.31 |
| FTFT | DeBERTaV3$_{Large}$ | DeBERTaV3$_{Base}$ | 79.76 | **82.05** | 76.77 | 78.62 | 75.22 | **78.43** | **71.00** |

Table 4: Comparison between FTFT and ERM/original DM fine-tuning. Performance of DeBERTaV3 on NLI (top, accuracy) and HSD (bottom, macro-F1). ERM is conventional ERM fine-tuning, and FTFT refers to using the 33% most ambiguous data identified by different reference models (i.e., Ref. Model) *and only training for 1/3 of the total steps*. Cost refers to the fine-tuning cost, with the cost of fine-tuning ELECTRA-Small with ERM as the unit. FTFT yields both better efficiency and better robustness compared to both ERM fine-tuning and the original DM method.

models, and compare FTFT against ERM, original DM, JTT (Liu et al., 2021), and PoE (Sanh et al., 2021). We also show the cost for each method for NLI, using the cost of fine-tuning ELECTRA$_{Small}$ with ERM as unit. The relative cost for each method in HSD is the same as NLI.

We make two observations. First, FTFT achieve better robustness than other methods, indicated by its strong performance on most challenging datasets. Second, fine-tuning models using FTFT is also more efficient than other methods. For example, FTFT with DeBERTaV3$_{Base}$ and DeBERTaV3$_{Small}$ as the reference models only cost 19.7 and 15.2 units of computation: compared with the 32/64 units of computation of ERM/DM fine-tuning, these two FTFT choices are respectively 1.63/3.26 times and 2.11/4.22 times cheaper.

# 6 CONCLUSIONS & LIMITATIONS

Fine-tuned PLMs has been shown vulnerable to OOD and adversarial inputs. The DM method has been shown to improve model robustness (Swayamdipta et al., 2020), however, it is computationally expensive. In this paper, we have presented a novel approach for fine-tuning PLMs, FTFT, which yields both better efficiency and better robustness over conventional ERM fine-tuning (§5). FTFT is built on the DM method, based on two observations: 1) reference model training dynamics are highly transferable across different model sizes (§4.1) and pretraining methods (§4.2), and 2) models trained using DM learn faster than when using conventional ERM fine-tuning. We have also discussed the conditions for successful FTFT runs (§4.3). We believe that FTFT will be an important tool for future researchers and practitioners to perform efficient PLM fine-tuning, especially in situations where robustness against out-of-distribution or adversarial inputs is essential.

Nevertheless, our work has limitations. First, although we observed that effective reference models identify more easy instances, we did not conduct controlled experiments to validate whether this feature is the only condition to ensure the success of transferring training dynamics. Future researchers can perform more exhaustive empirical studies to investigate how characteristics of reference models affect their effectiveness. Second, we have only tested FTFT on two classification tasks. Future studies can extend our work to tasks of other types, such as generative tasks (e.g., question answering), to examine the generalizability of FTFT.

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

# A  TRAINING SPECIFICATIONS

## A.1  EXPERIMENTAL SETUP

For training all models, we use AdamW (Loshchilov & Hutter, 2019) as the optimizer with a batch size of 32. We also use a linear learning rate scheduler with 10% warmup. For fine-tuning the small and base versions of both DeBERTa-V3 and ELECTRA, as well as TinyBERT, we use a learning rate of 2e-5. For DeBERTa-V3-Large and ELECTRA-Large, we respectively use 1e-5 and 2e-6 as the learning rates. We also experimented with 1e-5 and 5e-6 for ELECTRA-Large. However, we observed high ratios of failed runs, i.e., the training fails to converge and produces a worse than majority class baseline (Mosbach et al., 2021). We therefore follow the suggestions from Mosbach et al. (2021) to adopt a lower learning rate. We still encountered one failed run and we excluded that run in our results. For the PoE baseline, we use a cross entropy loss weight of 0.3 and a PoE loss weight of 1.0; for the JTT baseline, we used SGD optimizer, `reduce_lr_on_plateau` scheduler, and respectively the third and the fifth epoch as the reference epoch for NLI and HSD. We up-weigh four times for each mispredicted sample.

We use Python 3.9 and PyTorch 2.0 for all experiments. For training PLMs, we use HuggingFace Transformers 4.32 (Wolf et al., 2020), Accelerate 0.22, and Datasets 2.14 (Lhoest et al., 2021). All experiments are performed on two NVIDIA A100 GPUs. Training all models takes approximately seven GPU days.

## A.2  COMPARISON OF TRAINING COSTS

Here we show the training costs of different models (PFLOPs), of a single forward epoch. When taking back-propagation, optimization, and multiple epochs into consideration, these numbers should scale proportionally.

|  | NLI: MultiNLI | HSD: CAD |
|---|---|---|
| DeBERTaV3$_{Small}$ | 3116.93 | 312.35 |
| DeBERTaV3$_{Base}$ | 6233.34 | 624.68 |
| DeBERTaV3$_{Large}$ | 22160.84 | 2220.9 |
| ELECTRA$_{Small}$ | 694.61 | 69.98 |
| ELECTRA$_{Base}$ | 6227.14 | 627.37 |
| ELECTRA$_{Large}$ | 22138.79 | 2230.5 |
| TinyBERT | 28.88 | 2.91 |

Table 5: Comparison of training costs, measured in PFLOPs. Here we only calculate the forward cost for a single epoch.

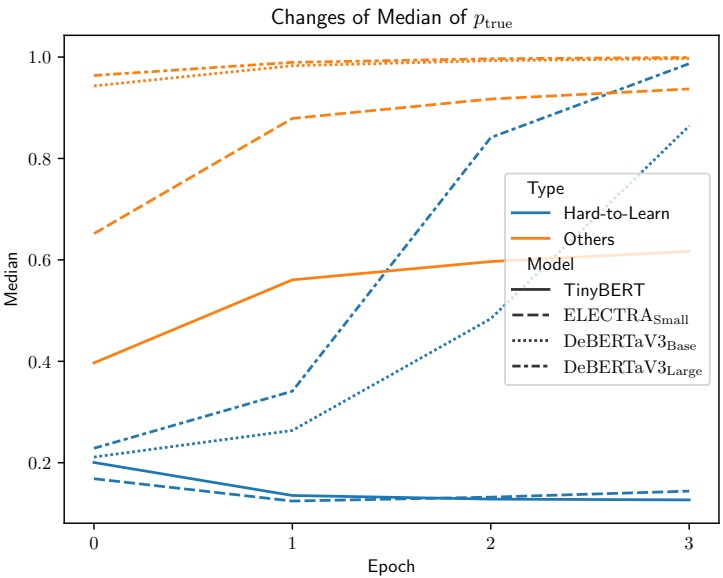

Figure 3: Change of Median $p_{\text{true}}$ over different training epochs

# B DIFFERENCES BETWEEN EFFECTIVE AND INEFFECTIVE REFERENCE MODELS

In §4.3, we reasoned about the differences between reference models that lead to successful and unsuccessful transfers of data maps. Our reasoning drew the conclusion that reasonably capable reference models that lead to successful transfers would 1) fit the easier training instances during the early stage of training and 2) fit the harder training instances gradually across different epochs; while too weak reference models would 1) fit the easier training instances gradually across different epochs and 2) fail to fit hard training instances (i.e. $p_{\text{true}}$ remains low during training). We then validated this reasoning by observing the percentage of overlaps between ambiguous and hard-to-learn training instances of different reference models.

As a complementary of our previously validation, here we offer more direct and even firmer evidence that our reasoning holds in practice. Specifically, for a given reference model, we take two steps. First, we split the training instances into two subsets according to their mean $p_{\text{true}}$, i.e., hard-to-learn (10% of all training instances) and others (90% of all training instances). We here use a lower $q\% = 10\%$ to make the difference clearer. Second, for each subset, we calculate the median $p_{\text{true}}$ in each epoch for these training instances. We use median values here because they are robust statistics of the central tendency. Concretely, we show our results on MultiNLI with four reference models, DeBERTaV3$_{\text{Large}}$, DeBERTaV3$_{\text{Base}}$, ELECTRA$_{\text{Small}}$, and TinyBERT. We use these four models because they are representative choices for effective and reasonably capable (DeBERTaV3$_{\text{Large}}$ and DeBERTaV3$_{\text{Base}}$), and ineffective and weak (ELECTRA$_{\text{Small}}$ and TinyBERT) reference models.

Our results are shown in Figure 3. It is clearly that for DeBERTaV3$_{\text{Large}}$ and DeBERTaV3$_{\text{Base}}$, hard-to-learn data instances are gradually learned over the training process, while other data instances already show high $p_{\text{true}}$ in the first epoch; while for ELECTRA$_{\text{Small}}$ and TinyBERT, hard-to-learn data instances are not learned at all, suggested by their close-to-zero $p_{\text{true}}$, and other data instances are gradually fitted by these two models. This result strongly support our reasoning from §4.3.

# C    ADDITIONAL RESULTS

| Mode | Main Model | Ref. Model | CAD | DynaHate-Original | | | DynaHate-Perturb | | |
| --- | --- | --- | --- | --- | --- | --- | --- | --- | --- |
| | | | | R2 | R3 | R4 | R2 | R3 | R4 |
| ERM | TinyBERT | - | 71.72 | 43.19 | 49.6 | 52.88 | 43.21 | 57.5 | 52.91 |
| ERM | ELECTRA$_{Small}$ | - | 74.15 | 48.93 | 56.93 | 58.93 | 55.75 | 61.84 | 59.52 |
| ERM | ELECTRA$_{Base}$ | - | 76.91 | 62.13 | 60.71 | 61.72 | 55.42 | 64.64 | 62.05 |
| ERM | ELECTRA$_{Large}$ | - | 68.07 | 70.07 | 62.56 | 70.27 | 63.91 | 70.45 | 67.02 |
| DM | DeBERTaV3$_{Large}$ | TinyBERT | 78.91 | 71.68 | 71.69 | 76.52 | 71.22 | 75.92 | 68.23 |
| DM | DeBERTaV3$_{Large}$ | ELECTRA$_{Small}$ | 79.74 | 78.09 | 77.4 | **78.75** | 75.26 | 76.79 | 70.05 |
| DM | DeBERTaV3$_{Large}$ | ELECTRA$_{Base}$ | **80.37** | **81.47** | **78.16** | 78.38 | **76.99** | **78.51** | **71.01** |
| DM | DeBERTaV3$_{Large}$ | ELECTRA$_{Large}$ | 79.48 | 76.71 | 75.97 | 78.58 | 73.55 | 77.8 | 69.81 |
| DM | ELECTRA$_{Large}$ | ELECTRA$_{Small}$ | 74.35 | 75.62 | 67.92 | 74.06 | 68.43 | 71.62 | 67.3 |
| DM | ELECTRA$_{Large}$ | ELECTRA$_{Base}$ | 75.84 | 75.52 | 65 | 72.89 | 67.94 | 70.34 | 67.94 |
| DM | ELECTRA$_{Large}$ | ELECTRA$_{Large}$ | 71.93 | 65.25 | 62.78 | 72.45 | 62.92 | 71.17 | 68.23 |
| DM | ELECTRA$_{Large}$ | Random | 74.14 | 62.14 | 58.13 | 68.23 | 63.3 | 67.93 | 65.22 |

Table 6: Performance of different models on HSD with conventional fine-tuning (ERM) and DM using the 33% most ambiguous data identified with different reference models. Random in Ref. Model means randomly selecting 33% of the train data. The rows marked in gray are the results for DM training where the transfer was not successful, and the corresponding reference models. Successful transfer requires the reference model to be reasonably strong: reference models of clearly worse performance lead to degraded OOD performance for the main models. Note that although using TinyBERT and DeBERTaV3$_{Large}$ respectively as the reference and the main model yields better performance than ELECTRA, that is due to the better performance of DeBERTaV3$_{Large}$ than ELECTRA$_{Large}$: it is still worse than other methods using ELECTRA$_{Large}$ as the main model, especially on OOD test sets.

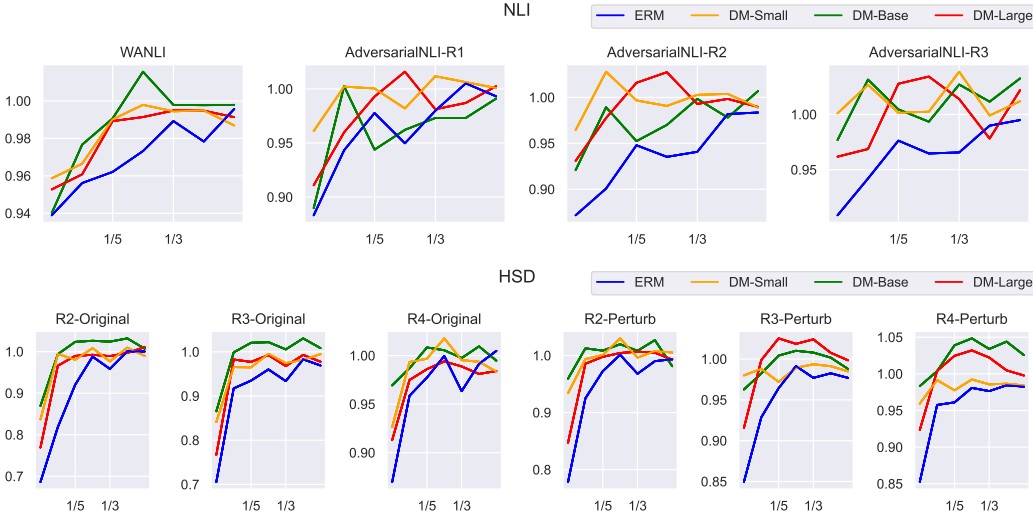

Figure 4: Performance on NLI (top) and HSD (bottom) when training the main model (DeBERTaV3) with fewer training steps. ERM is standard ERM fine-tuning on the full training set. DM-* refers to fine-tuning ELECTRA$_{Large}$ with the DM method, using reference model ELECTRA$_*$. The X-axis is the percentile of full training steps used, ranging from 1/15 to 7/15 of the total number of training steps. The Y-axis is the **percentile of performance compared with the full training steps**. Fine-tuning the main model using data maps is much faster than ERM: the models achieve close-to-100% performance using only 1/3 of the training steps.

| Mode | Main Model | Ref. Model | Cost | MultiNLI | WANLI | AdversarialNLI | | |
|------|-----------|-----------|------|----------|-------|------|------|------|
| | | | | | | R1 | R2 | R3 |
| ERM | ELECTRA$_{Large}$ | - | 32.0 | 90.75 | 65.85 | 54.2 | 39.38 | 36.1 |
| DM | ELECTRA$_{Large}$ | ELECTRA$_{Large}$ | 64.0 | 90.33 | 65.37 | 53.73 | 39.67 | 36.17 |
| FTFT | ELECTRA$_{Large}$ | ELECTRA$_{Small}$ | 11.7 | 89.46 | 60.87 | 46.05 | 35.82 | 32.6 |
| FTFT | ELECTRA$_{Large}$ | ELECTRA$_{Base}$ | 19.7 | **90.06** | **65.85** | **54.7** | **40.23** | **37.58** |

| Mode | Main Model | Ref. Model | CAD | DynaHate-Original | | | DynaHate-Perturb | | |
|------|-----------|-----------|-----|------|------|------|------|------|------|
| | | | | R2 | R3 | R4 | R2 | R3 | R4 |
| ERM | ELECTRA$_{Large}$ | - | 68.07 | 70.07 | 62.56 | 70.27 | 63.91 | 70.45 | 67.02 |
| DM | ELECTRA$_{Large}$ | ELECTRA$_{Large}$ | 71.93 | 65.25 | 62.78 | 72.45 | 62.92 | 71.17 | 68.23 |
| FTFT | ELECTRA$_{Large}$ | ELECTRA$_{Small}$ | 75.52 | 73.45 | **66.54** | **73.79** | 67.08 | **72.05** | 68.11 |
| FTFT | ELECTRA$_{Large}$ | ELECTRA$_{Base}$ | **77.15** | **74.17** | 65.53 | 73.42 | **68.77** | 70.6 | **68.7** |

Table 7: Comparison between FTFT and ERM/original DM fine-tuning. Performance of ELECTRA on NLI (top, accuracy) and HSD (bottom, macro-F1). ERM is conventional ERM fine-tuning, and FTFT refers to using the 33% most ambiguous data identified by different reference models (i.e., Ref. Model). Cost refers to the fine-tuning cost, with the cost of fine-tuning ELECTRA-Small with ERM as the unit. FTFT yields both better efficiency and better robustness compared to both ERM fine-tuning and the original DM method. For NLI, we only train the main model for 1/3 of the total steps. For HSD, we observe that ELECTRA converges a bit slower, see Figure 2. We therefore use 7/15 of the total steps for FTFT to obtain better performance (still less than 1/2), making the cost 15.9 and 23.9 for small and base reference models respectively. FTFT yields both better efficiency and better robustness, compared to both ERM fine-tuning and the original DM method, except for using ELECTRA$_{Small}$ as the reference model on NLI.

