# OpenReview forum: "FTFT: efficient and robust Fine-Tuning by transFerring Training dynamics"
_ICLR.cc/2024/Conference — ICLR 2024 Conference Withdrawn Submission_

### Official Review · Reviewer_dPV5 · 2023-10-30

**Soundness:** 3 good
**Presentation:** 3 good
**Contribution:** 1 poor
**Rating:** 3
**Confidence:** 3

**Summary:**

This paper demonstrates that (1) smaller models can also be used as reference models in data map (DM) methods as described in Swayamdipta et al. 2020 and (2) transfer is possible between models with different pre-training methods on NLI and hate speech detection benchmarks. They also conduct ablation studies on how fast models learn with ERM vs DM. I cannot recommend this paper for acceptance as the novelty is very limited over Swayamdipta et al. 2020.

**Strengths:**

Although the novelty is limited, ablations and benchmarking shown are quite detailed. Presentation is very clear. Related literature is very well-reviewed.

**Weaknesses:**

Contribution is extremely limited over Swayamdipta et al. (2020) who proposed the original method of data maps combined with the work of Sar-Shalom & Schwartz (2023) who demonstrated that a DM constructed by ELECTRALarge can be used to improve the robustness of DeBERTaV3Large. This work reads more like a tech report rather than an ICLR paper. Insights are practically useful, but does not go beyond systematic benchmarking.

**Questions:**

I do not have any questions -- presentation is clear.

---

> ### Author Response · Authors · 2023-11-16
> **Author Response to Reviewer dPV5**
>
> Thank you for your comments.
>
> > Contribution is extremely limited over Swayamdipta et al. (2020) … combined with the work of Sar-Shalom & Schwartz (2023) ... but does not go beyond systematic benchmarking
>
> We agree that our approach builds upon the foundational concepts introduced in the original data map paper, and takes inspiration from Sar-Shalom & Schwartz (2023). However, in this paper, we investigate a **distinct and unexplored research question**: to which extent data maps are transferable across different model sizes and pretraining methods. Addressing this question can open up a range of new applications (like our approach), and deepen our understanding of the relationship between model architecture and the importance of specific training data instances. To this end, we systematically examine the transferability of data maps. Our results indicate promising transferability, and we use this finding to improve both robustness and training efficiency by proposing a novel fine-tuning approach: FTFT.
>
> **Moreover, our work differs substantially with Sar-Shalom & Schwartz (2023)** in that 1) Sar-Shalom & Schwartz (2023) adopted a different and less efficient algorithm based on data maps, which increases the complexity of the original data map algorithm, let alone ERM fine-tuning; 2) Sar-Shalom & Schwartz (2023) did not consider or study the implication for efficiency for such transfers. They experimented with two **equally expensive** models and did not go further to study the specific transferability of data maps across different factors.

---

### Official Review · Reviewer_CV2g · 2023-10-31

**Soundness:** 3 good
**Presentation:** 2 fair
**Contribution:** 2 fair
**Rating:** 5
**Confidence:** 4

**Summary:**

The authors propose a method for finetuning pretrained language models, FTFT, that finetunes a smaller reference model which is then used to select examples for training the target model on a downstream task. The authors demonstrate that smaller models can be used for constructing a DataMap of samples without significant reductions in performance.

**Strengths:**

1. The authors conduct a systematic investigation of various reference model sizes and compare against reference models trained with an alternative discriminative pretraining method.
2. The authors demonstrate that using smaller reference models and training on the resulting DataMap does not result in performance reduction as compared with an ERM baseline trained over the entire dataset; and results in improved performance on OOD robustness datasets.

**Weaknesses:**

1. One of the primary contributions is the sample efficiency of models trained on a smaller data map selected via ambiguity.  However, there is limited comparison or discussion of related work on sample efficient methods of training such as curriculum learning and dataset pruning [1, 2] .
2. The DataMap selection criteria is limited to example ambiguity -- and does not compare against other criteria such as "hard-to-learn", example forgetability [3]
3. Evaluations are limited to finetuning of models for language classification -- unclear whether results would generalize to other domains or task settings (e.g. image classification, language generation).

References:
1. Sorscher, Ben, et al. "Beyond neural scaling laws: beating power law scaling via data pruning." Advances in Neural Information Processing Systems 35 (2022): 19523-19536.
2. Paul, Mansheej, Surya Ganguli, and Gintare Karolina Dziugaite. "Deep learning on a data diet: Finding important examples early in training." Advances in Neural Information Processing Systems 34 (2021): 20596-20607.
3. An empirical study of example forgetting during deep neural network learning. In ICLR, 2019.

**Questions:**

* Significance of performance gains over baseline DataMap are unclear without variance across random seeds in Tables {1, 2, 4}?
* Table 4; How is the cost of one "ELECTRA-Small with ERM" calculated (i.e. FLOPs, GPU-Hours, power consumption?) Does this account for the cost of finetuning the reference model and scoring the samples DataMap?
* Why is 33% chosen for the top q% to create the Data Map? What is the distribution of the ambiguous and hard to learn examples? Do the values have a large degree of skewness?

---

> ### Author Response · Authors · 2023-11-16
> **Author Response to Reviewer CV2g (1/2)**
>
> Thank you for your helpful comments and questions.
>
> > However, there is limited comparison or discussion of related work on sample efficient methods of training such as curriculum learning and dataset pruning [1, 2] .
> > The DataMap selection criteria is limited to example ambiguity -- and does not compare against other criteria such as "hard-to-learn", example forgetability [3]
>
> **Curriculum learning and dataset pruning methods**: We appreciate your suggestion to include these topics. We will incorporate discussions on curriculum learning and dataset pruning methods within the related work section. However, it's important to note that our research primarily emphasizes robustness against challenging OOD data. Therefore, there is a less direct relationship between these approaches and FTFT.
>
> **Hard-to-learn and example forgettability**: Swayamdipta et al. (2020) observe that training using ambiguous data yields better performance than using hard-to-learn and forgettable instances. We therefore did not compare against these two methods, since FTFT outperforms the original data map method using ambiguous data. However, we recently experimented with two other robustness enhancement training approaches, PoE [1] and JTT [2]: we found that, compared with these two approaches, FTFT achieves comparable or better performance despite using less-than-half of the training budget. See Table 4 in our updated version.
>
> > Evaluations are limited to finetuning of models for language classification -- unclear whether results would generalize to other domains or task settings (e.g. image classification, language generation).
>
> In this paper, we focus on classification tasks because the data map method was proposed to address text classification robustness. It is particularly relevant to text classification in view of the known susceptibility of language models to spurious correlation and bias. Nevertheless, data maps can be straightforwardly applied to many classification tasks, including image classification, and we hope our work serves to inspire such work in the CV community.
>
> However, because the data map method was not developed for generation tasks, it is not straightforward to perform language generation evaluation. Nevertheless, we believe developing generation data maps is a promising direction to look at for future studies.
>
> > Table 4; How is the cost of one "ELECTRA-Small with ERM" calculated (i.e. FLOPs, GPU-Hours, power consumption?) Does this account for the cost of finetuning the reference model and scoring the samples DataMap?
>
> We estimate the cost for training each model by calculating the FLOPs, using https://github.com/sovrasov/flops-counter.pytorch/tree/master. **Our cost estimation for each method considers the training of both reference models and main models**.
>
> >> Why is 33% chosen for the top q% to create the Data Map? What is the distribution of the ambiguous and hard to learn examples? Do the values have a large degree of skewness?
>
> We use q%=33% to follow the choices of Swayamdipta et al. (2020). In their experiments, they demonstrate that using less ambiguous data will make models less likely to converge; and using more examples will make FTFT less efficient.
>
> [1] Learning from others' mistakes: Avoiding dataset biases without modeling them. Sanh et al., 2021.
>
> [2] Just Train Twice: Improving Group Robustness without Training Group Information. Liu et al., 2021.

---

> ### Author Response · Authors · 2023-11-16
> **Author Response to Reviewer CV2g (2/2)**
>
> > Significance of performance gains over baseline DataMap are unclear without variance across random seeds in Tables {1, 2, 4}?
>
> We show the standard deviation of results of DeBERTaV3 using ERM and reference models of different sizes here, and will provide standard deviations for each experiment in our appendices. We can see that the standard deviation of using different sizes of reference models are comparable.
>
> **NLI**
>
> | Mode | Main Model                        | Ref. Model                        | MultiNLI   | WANLI      | AdversarialNLI |            |            |
> |------|-----------------------------------|-----------------------------------|------------|------------|----------------|------------|------------|
> |      |                                   |                                   |            |            | R1             | R2         | R3         |
> | ERM  | $\text{DeBERTaV3}_{\text{Small}}$ | -                                 | 87.57±0.08 | 61.62±0.12 | 33.55±1.23     | 30.63±0.89 | 32.40±0.77 |
> | ERM  | $\text{DeBERTaV3}_{\text{Base}} $ | -                                 | 90.01±0.12 | 64.61±0.28 | 43.55±0.68     | 33.23±1.08 | 34.15±0.40 |
> | ERM  | $\text{DeBERTaV3}_{\text{Large}}$ | -                                 | 91.06±0.08 | 66.46±0.33 | 58.18±1.50     | 45.58±0.64 | 41.34±1.12 |
> | DM   | $\text{DeBERTaV3}_{\text{Large}}$ | Random                            | 90.74±0.14 | 65.31±0.78 | 53.30±2.33     | 42.03±1.36 | 38.60±1.17 |
> | DM   | $\text{DeBERTaV3}_{\text{Large}}$ | $\text{DeBERTaV3}_{\text{Large}}$ | 90.75±0.29 | 66.33±0.07 | 59.75±0.86     | 45.60±1.86 | 41.94±0.80 |
> | DM   | $\text{DeBERTaV3}_{\text{Large}}$ | $\text{DeBERTaV3}_{\text{Small}}$ | 90.74±0.21 | 66.80±0.61 | 59.60±1.14     | 45.63±1.12 | 42.04±0.66 |
> | DM   | $\text{DeBERTaV3}_{\text{Large}}$ | $\text{DeBERTaV3}_{\text{Base}}$  | 90.52±0.05 | 66.61±0.76 | 61.43±1.44     | 46.73±0.92 | 41.58±0.94 |
>
> **HSD**
>
> | Mode | Main Model                        | Ref. Model                        | CAD        | DynaHate-Original |            |            | DynaHate-Perturb |            |            |
> |------|-----------------------------------|-----------------------------------|------------|-------------------|------------|------------|------------------|------------|------------|
> |      |                                   |                                   |            | R2                | R3         | R4         | R2               | R3         | R4         |
> | ERM  | $\text{DeBERTaV3}_{\text{Small}}$ | -                                 | 76.58±0.74 | 56.89±5.13        | 59.29±3.89 | 63.48±0.99 | 59.55±2.67       | 66.60±1.51 | 61.48±1.30 |
> | ERM  | $\text{DeBERTaV3}_{\text{Base}} $ | -                                 | 78.64±0.55 | 60.53±2.22        | 64.28±0.83 | 68.89±2.00 | 60.81±1.56       | 69.48±1.30 | 63.12±2.02 |
> | ERM  | $\text{DeBERTaV3}_{\text{Large}}$ | -                                 | 81.69±0.57 | 75.44±1.67        | 73.33±0.80 | 76.12±1.54 | 70.62±1.83       | 77.41±0.57 | 68.89±1.01 |
> | DM   | $\text{DeBERTaV3}_{\text{Large}}$ | Random                            | 76.23±0.98 | 63.38±2.00        | 61.59±3.48 | 71.21±2.36 | 64.05±1.78       | 72.10±1.45 | 62.88±2.24 |
> | DM   | $\text{DeBERTaV3}_{\text{Large}}$ | $\text{DeBERTaV3}_{\text{Large}}$ | 81.58±0.72 | 79.18±1.12        | 76.87±1.89 | 77.73±1.42 | 73.35±1.14       | 76.63±0.94 | 67.54±0.50 |
> | DM   | $\text{DeBERTaV3}_{\text{Large}}$ | $\text{DeBERTaV3}_{\text{Small}}$ | 81.15±0.33 | 80.68±3.14        | 79.57±0.64 | 79.86±1.60 | 76.47±0.98       | 78.03±0.16 | 70.32±0.94 |
> | DM   | $\text{DeBERTaV3}_{\text{Large}}$ | $\text{DeBERTaV3}_{\text{Base}}$  | 80.12±0.76 | 80.13±1.21        | 76.35±3.41 | 78.82±0.78 | 74.60±1.30       | 77.82±1.13 | 68.68±0.54 |

---

### Official Review · Reviewer_FKcX · 2023-10-31

**Soundness:** 3 good
**Presentation:** 3 good
**Contribution:** 3 good
**Rating:** 5
**Confidence:** 3

**Summary:**

The paper proposes FTFT, an efficient fine-tuning algorithm that selects a core set of examples to fine-tune a large model by using the training dynamics of a small reference model. The authors observe that such an algorithm can achieve better OOD performance with a slight drop in ID performance when compared to the conventional ERM algorithm. The authors conduct extensive experiments to find the right reference model to select the core set, where the selection can be made based on model size and family. Finally, the authors show the efficiency gains of their method compared to ERM by comparing the behavior of the model's OOD performance over training time.

**Strengths:**

The strength of the paper lies in its easy-to-understand explanation of the algorithm. The authors begin with a clear description of the existing literature on the data map methods and the underlying issue of these methods. With a proposed hypothesis of using a small model to provide the necessary data map, the authors test multiple candidates that can act as the reference small model. Finally, extensive experimentation shows the efficacy of their method on multiple ID-OOD dataset pairs.

**Weaknesses:**

I have a few questions regarding the experimental setup.

(a) How efficient is FTFT compared to ERM in terms of total flops? Since FTFT first trains a small reference model to select the ambiguous set of examples, it has to incur the flop necessities of training the small reference model. A rough estimate of the flop counts for both methods will be useful.

(b)  How does FTFT perform when compared to existing algorithms that aim to improve the OOD performance of trained models? Examples of such methods include invariant risk minimization algorithms [4], DRO [5], and WiSE-FT [6]. Comparison to a couple of them will strengthen the results of the FTFT method.

(c) How sensitive is FTFT to training hyperparameters of the small reference model and the target model? Does the ambiguous core set selected using the small reference model change with its training hyperparameters?

(d) I observed that the core set selected with base models (ELECTRA-base and DeBERTaV3-base) performs better OOD than training with a core set selection from large models. Can the authors provide more insights into the behavior?


There are a number of papers that I believe should be part of the related works section to give readers a full overview of the literature.
For example, [1] dynamically weighs training domains in the pre-training dataset of a large language model, using the training dynamics of a small language model. Other citations may include works that train a proxy model to select the right set of data to train the target model. [2, 3]


1: DoReMi: Optimizing Data Mixtures Speeds Up Language Model Pretraining. Xie et al.' 23

2: Selection via proxy: Efficient data selection for deep learning. Coleman et al.' 19

3: SVP-CF: Selection via Proxy for Collaborative Filtering Data. Sachdeva et al.'21

4: Invariant Language Modeling. Peyrard et al.'21

5:  Distributionally robust language modeling. Oren et al.'19

6: Robust fine-tuning of zero-shot models. Wortsman et al'21

**Questions:**

Please see my questions in the previous section.

---

> ### Author Response · Authors · 2023-11-16
> **Author Response to Reviewer FKcX**
>
> Thank you for your helpful comments and questions.
>
> > How efficient is FTFT compared to ERM in terms of total flops?
>
> Our training cost estimation for each approach, including FTFT and ERM, is calculated by the total FLOPs (i.e. **including the training of both reference models and main models**), using https://github.com/sovrasov/flops-counter.pytorch/tree/master. FTFT costs less-than-half of the training budget of ERM considering both phrases of training.
>
> > How does FTFT perform when compared to existing algorithms that aim to improve the OOD performance of trained models?
>
> Thank you for pointing this out. We agree it is important to include comparisons with other methods for improving model robustness. However, IRM, DRO, and WiSE-FT work with splits of distributions according to some known data properties (e.g. topics, genres), while data map and FTFT are distribution-agnostic approaches, and can work with any classification dataset without further curation.
>
> However, there do exist other distribution-agnostic approaches, e.g. PoE [1] and JTT [2]. Our comparison with these two methods shows that FTFT achieves comparable or better performance despite using less-than-half of the training budget. See Table 4 in our updated version.
>
> > How sensitive is FTFT to training hyperparameters of the small reference model and the target model?
>
> We agree that it is important to ensure the robustness of FTFT towards hyper-parameter selections. However, we note that in our experiments we use the default hyper-parameters recommended in the original DeBERTaV3 and ELECTRA papers without further tuning, and we simply tie the hyper-parameters of reference models and main models. We thus expect our results to be a representative demonstration of FTFT.
>
> >  I observed that the core set selected with base models (ELECTRA-base and DeBERTaV3-base) performs better OOD than training with a core set selection from large models. Can the authors provide more insights into the behavior?
>
> Thank you for this intriguing question! We suspect that large models fit all data instances well at the early stage of training, making ambiguous data and easy data less distinguishable. We encourage future studies to perform further analysis on the differences between the specific ambiguous data instances identified by reference models with different capabilities.
>
> > There are a number of papers ... For example, [1] dynamically weighs training domains … train a proxy model to select the right set of data to train the target model. [2, 3]
>
> Thank you for the references. We agree that including these papers will strengthen our related work section, and will for sure include these discussions in our next version.
>
> [1] Learning from others' mistakes: Avoiding dataset biases without modeling them. Sanh et al., 2021.
>
> [2] Just Train Twice: Improving Group Robustness without Training Group Information. Liu et al., 2021.

---

### Official Review · Reviewer_vdXa · 2023-11-01

**Soundness:** 3 good
**Presentation:** 2 fair
**Contribution:** 1 poor
**Rating:** 3
**Confidence:** 4

**Summary:**

To reduce the training cost of Data map. This paper develops a variant of Data map by swapping in a smaller model for data selection. The authors experimented on DEBERTA, ELECTRA and TinyBERT with a few datasets, showing training cost improvement and mixed results in performance.

**Strengths:**

1. Simple and clear idea.
2. The motivation and reasoning are well explained.

**Weaknesses:**

1. The proposed method has limited novelty compared to Data map.
2. The result is mixed. Out of the experiments in Table 2, only half of them show successful transfer. Suggesting the scale of the reference model still needs to be relatively close to the Main model. Besides, even in the remaining rows, the result is inconsistent across datasets.
3. Cost saving is only in fine-tuning time rather than inference time. However, for LLMs, fine-tuning cost is much less of an issue than pertaining or inference cost.

**Questions:**

N/A

---

> ### Author Response · Authors · 2023-11-16
> **Author Response to Reviewer vdXa**
>
> Thank you for your comments.
>
> > The proposed method has limited novelty compared to Data map.
>
> We agree that our approach builds upon the foundational concepts introduced in the original data map paper. However, in this paper, we investigate a **distinct and unexplored research question**: to which extent data maps are transferable across different model sizes and pretraining methods. Addressing this question can open up a range of new applications (like our approach), and deepen our understanding of the relationship between model architecture and the importance of specific training data instances. To this end, we systematically examine the transferability of data maps. Our results indicate promising transferability, and we use this finding to improve both robustness and training efficiency by proposing a novel fine-tuning approach: FTFT.
>
> > The result is mixed.
>
> To study the transferability of data maps, we deliberately also included weak reference models (e.g., TinyBERT) that would more likely result in failed transfers. Including such models allowed us to analyze the conditions for successful transfers. Table 2 highlights the situations where the transfer of data maps from reference models to main models fails. If we only consider the main models that we experimented with in this paper (DeBERTaV3 and ELECTRA of three different sizes), only 1 out of 6 models fail for NLI, and no models fail for HSD. To further investigate such transferability, we further experimented with BERT-Large and RoBERTa-Large as reference models for DeBERTaV3-Large, and our results again suggest good transferability (See Table 1 in our updated version). In our experiments, we show that by choosing smaller reference models we can improve training efficiency by at least 2x, while showing better robustness.
>
> In Section 4.3 we analyze why very weak reference models lead to failed transfers and offer insights for selecting effective yet efficient reference models. To further validate our findings in Section 4.3, i.e., weak reference models tend to identify easy data as ambiguous data, we have performed the following additional experiment with four models, TinyBERT, ELECTRA-Small, DeBERTaV3-Base, and DeBERTaV3-Large (see Figure 3 in Appendix B). First, for each model, we separate the training dataset of MultiNLI into two groups, the 10% most hard-to-learn data and the remaining (i.e. easier) data. Second, we plot the median value of p_true for each data group. Our results show that for models that fail to transfer their data maps (i.e., ELECTRA-Small and TinyBERT), hard-to-learn data instances have consistent low p_true values across epochs, and consequently receive lower ambiguous scores. In this case, the data instances that are identified as ambiguous are actually the easier ones.  However, for DeBERTaV3-Base/Large, which successfully transfer their data maps, easier data converges at the very beginning of the training process, and thus also receive lower ambiguous scores.
> “Even in the remaining rows, the result is inconsistent across datasets.” We are not sure which results you are referring to here: compared with the original data map method, we observe comparable or better results in successful transfers.
>
> > Cost saving is only in fine-tuning time rather than inference time. However, for LLMs, fine-tuning cost is much less of an issue than pertaining or inference cost.
>
> As fine-tuning still outperforms in-context learning in NLP tasks [1], its efficiency and robustness is of great importance. In this work, we simultaneously improve both fine-tuning efficiency and prediction robustness. We believe our work is still of significance to modern NLP research.
>
>
> [1] Few-shot Fine-tuning vs. In-context Learning: A Fair Comparison and Evaluation. Mosbach et al, 2023.